# Spring-Damped Underactuated Swashplateless Rotor on a Bicopter Unmanned Aerial Vehicle

Haofei Guan [ID] and K. C. Wong *[ID]

School of Aerospace, Mechanical and Mechatronic Engineering, The University of Sydney, NSW 2006, Australia; hgua8006@uni.sydney.edu.au
* Correspondence: kc.wong@sydney.edu.au

**Abstract:** The stabilisation capabilities of unmanned aerial vehicles (UAVs) with bicopter underactuated swashplateless rotors are highly sensitive to motor-induced vibration. Due to the requirement of the active control of underactuated swashplateless rotors, conventional designs are limited in reducing vibration through control optimisation. A solution with customized passive spring-damping structures on a unique underactuated swashplateless rotor of a tiltrotor bicopter platform is presented. The implementation of this structure effectively reduces the self-coherent vibration in flights. As a result, a higher level of control authority has been achieved without setting excessive low-pass filtering for vibration. Experimentally obtained inertial measurement unit (IMU) data, rotor speed, rotor tilt angle, and the cyclic stator response are presented for comparison with Simulink model predictions.

**Keywords:** underactuation; vibration; rotors; propellers; blades; torque; tiltrotor; actuators





## 1. Introduction

Research on underactuated, swashplateless unmanned aerial vehicles (UAVs) has gradually emerged in recent years. This is mainly because conventional UAV formats for smaller micro air vehicles have posed significant design challenges. A study has prompted the exploration of underactuated rotors for simple micro air vehicles to achieve thrust, roll, pitch, and yaw from only two motors [1]. One study specifically focused on the flight performance of their swashplateless micro air vehicle, highlighting the challenges and opportunities associated with underactuated models [2]. Recently, underactuated swashplateless mechanism applications have been explored in self-rotating micro air vehicles, utilising the same principles on both thrust and moment [3]. The development of single-rotor UAV swashplateless torque modulation with various capabilities has been studied, highlighting the unique characteristics of highly underactuated UAVs [3]. Modelling and scalability research on the conventional design has also been proposed [4,5]. Overall, the existing research suggests a growing interest in underactuated swashplateless UAVs, focusing on addressing the design challenges associated with smaller or micro aerial vehicles and exploring new control methods for high-speed rotation and simultaneous localisation and mapping.

Vibration reduction is pivotal in enhancing unmanned aerial vehicles' (UAVs) performance, stability, and longevity. Besides being fully functional, vibration due to implementing torque-modulated underactuated swashplateless mechanisms is also inevitably induced into platform structures and avionics in all current state-of-the-art underactuated swashplateless rotor designs. Gyroscopic components in a vibratory system such as an aircraft was proposed as a topic in the 1940s. It is recognized that the precession-permitted and gyroscopic action contributes to a force that is opposed to the tilt and proportional to the angle of tilt [6]. Precession is also permitted in the underactuated propeller design, as constrained three-degree-of-freedom (C3DoF) underactuated rotating propellers have two free DoFs through the rotating linkages. Most importantly, this effect on underactuated

swashplateless propellers is neither the same as rigid body and primary rotor modelling on helicopters [7] nor utterly the same as the coupled rigid propeller engine gyroscopic model [8], as it does not have the free two DoFs on a rotating propeller. Without introducing extra structures, gyroscopic responses to disturbance indicate that separate stabilizing feedback controls are required [9].

On existing successful flight platforms, minor angular disturbances induced by this effect might have a limited impact on lift vectoring capability when the axial drive consistently stays in the centre of the platform [2,10]. On underactuated bicopters or even quadcopters, given the use of torque-modulated actuators for platform control, control-induced free-body vibrations on UAVs also results in the minor radially shaking motion of the propeller pivot point on each side, which, as expected, induces gyroscopic effects on the underactuated rotor disk. Unfortunately, in this case, the modulation signal is already used as the source of rotor underactuation control and is, thereby, the source of vibration, making it challenging to implement any similar input control methods to stabilize the rotation disturbance, as mentioned [9].

Since precession and nutation occur on a pivoted gyroscope, this effect impacts vectoring control accuracy. As a result, it is expected that the control-coherent vibration due to cyclic torque modulation consistently reduces the accuracy of the rotor vector controlling for a stable flight when the actuator is placed away from the vehicle centre axis. However, removing vibrations by turning off the torque modulation is also not an option, as it is identical to removing the vectoring DoF and losing control.

However, this paper provides simulation and experimental results showing that a spring-damping structure can significantly reduce control-inherent vibrating loads. Moreover, the primary issue this research solves is alleviating this effect by reducing the control-inherent vibration being transferred into structures. The results of this study show that a spring-damping structure is effective in countering control-inherent vibration.

This article explores the significance of vibration mitigation techniques in the context of an innovative, fixed blade pitch, underactuated swashplateless mechanism on a UAV platform. Excessive vibration can degrade flight stability, compromise payload integrity, and accelerate component wear, posing significant challenges to mission success and platform reliability. As proposed in this article, underactuated swashplateless UAVs can achieve smoother hovering operations by implementing passive damping materials. This article highlights the critical role of vibration reduction in unlocking the full potential of underactuated swashplateless UAVs with improved performance, reliability, and safety.

It is also expected that tiltrotor mechanisms will be explored in depth for aircrafts of different sizes, varying from micro aerial vehicles (MAVs) to electric vertical-take-off-and-landing (eVToL) vehicles, in the future. Both tiltwing and tiltrotor designs increasingly attract research attention [11]. Regarding hybrid UAV design and construction, the underactuated propeller with two degrees of freedom (DoFs) of self-driven thrust vectoring capability is worth exploring to identify predictable difficulties and corresponding solutions. Thereby, the outcome of this study also highlights several optional directions and hints for future scalability research.

## 2. Proposed Design

Our proposed design in this study is a passive spring-damping structure mounted between the platform and motor stator while bypassing the rotary encoder that reads the shaft angle. Unlike conventional underactuated swashplateless rotor platforms, this passive spring-damping structure is implemented on a bicopter platform utilising a C3DoF mechanism for smoother vectoring control. However, similar to conventional underactuated swashplateless rotor control, the torque pulse-driving waveform is also sinusoidal ($\psi_{rotor}-\omega$ sine wave control) [10]. Figure 1 provides an overview of the platform's layout being tested in this study. Meanwhile, the platform stabilises in tiltrotor mode, as shown in Figure 2, as the thrust vectoring control does not aim to induce cyclic blade pitch. In the Kakute F4 mini Ardupilot flight controller settings, this platform uses Ardupilot FixedWing 4.4.4 tail-sitter

mode as the firmware. The utilisation of the stabilisation method requires a test rig-tuned cascade PID flight controller on the platform without lateral tilt-vectoring capabilities. The average rotor rotation speed (bias) determines the average thrust generated by each rotor. The cyclic acceleration and deceleration magnitude of rotor speed (amplitude) determines the rotor thrust vectoring. The cyclic phase angle (phase) determines the thrust vectoring angle of the rotor. In the flight controller firmware, by default, differential thrust is used to stabilise the platform roll axis. The stabilisation method of the platform with a linearised control model is similar to conventional tiltrotor bicopter UAV PID control models [12,13] tuned prior to free flights. Although in further flight tests, this can also be replaced with thrust vectoring roll motion stabilisation to avoid excessive differential thrust-induced yaw effects, only the baseline stabilization method is required to observe the results in this study. Since the dual rotors rotate in opposite directions, the phase angles are calibrated separately and mirrored in phase angle calculations to obtain the correct vector tilting angle. An overview of the avionics communication, including the ground control station (GCS), is provided in Figure 3.

An onboard ESP32 MCU is assigned the following tasks:

- Generating sinusoidal pulse control signal to ESC using Oneshot125 protocol at 4 kHz (task 1, on MCU core 1).
- Retrieve the 14-bit rotary encoder data with SPI communication with a stable clock following task 1 (task 1, on MCU core 1, SPI protocol).
- Passing the telemetry signal with the rest of the clock time using the MAVLink protocol with a refresh rate of 100 Hz, parallel to other tasks (task 0, on MCU core 0).
- Retrieving PWM signal from flight controller varying from 50 Hz to 490 Hz, parallel to other tasks (digital-pin-change interruption).

Since maintaining communication with ESC at a desirably high frequency directly relates to the quality of the platform rotor vectoring control, this command is accomplished by assigning a separate core of a dual-core ESP32 MCU. Under highly compromised conditions, operating the ESC control frequency at 1 kHz (wholly occupying 75% of clock speed on other tasks), a propeller operating frequency at 50 Hz in hover remains at a minimum of 20 control access per revolution, which is high enough to define sinusoidal pulse amplification.

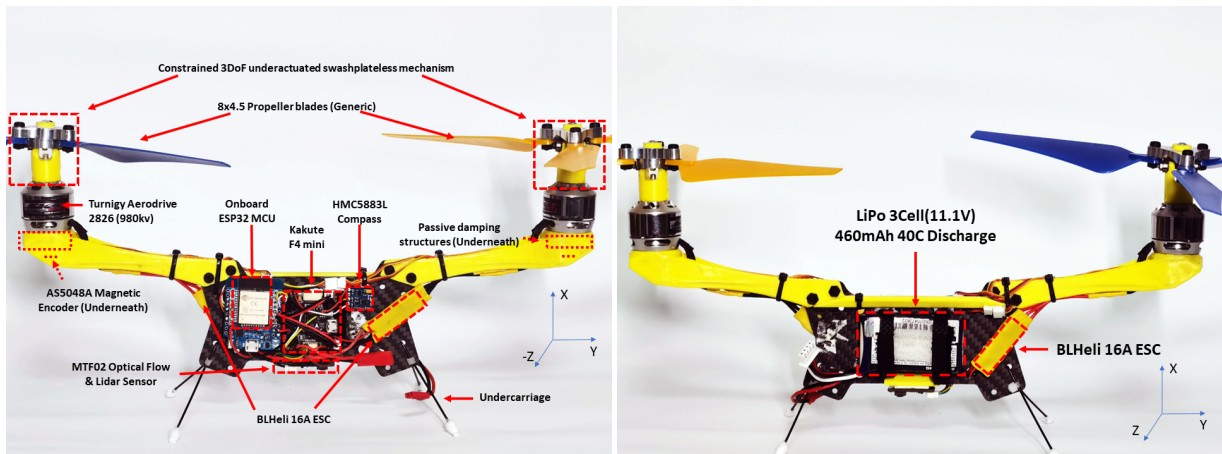

**Figure 1.** Platform overview of a constrained 3DoF underactuated swashplateless bicopter. (**left**) The front view of the platform. (**right**) The rear view of the platform.

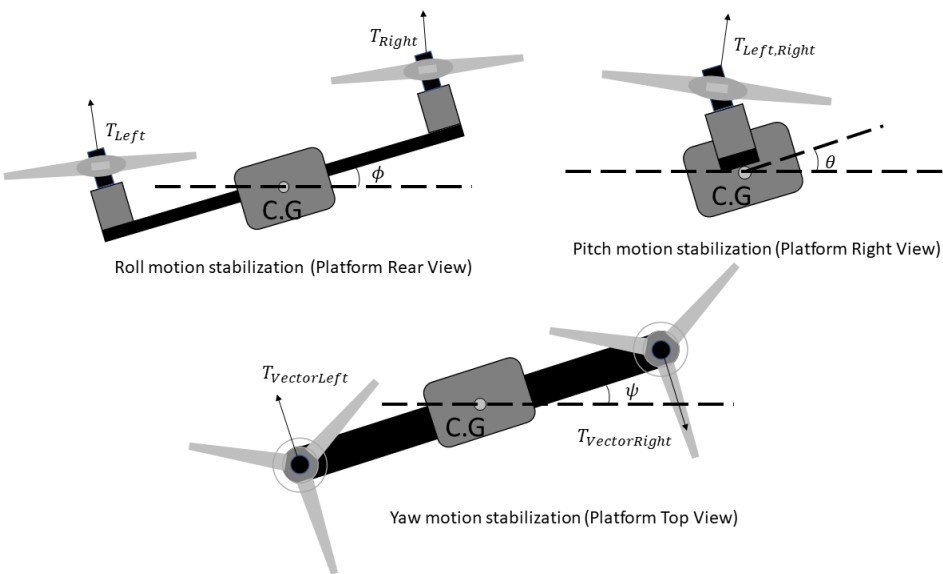

**Figure 2.** Platform stabilisation method.

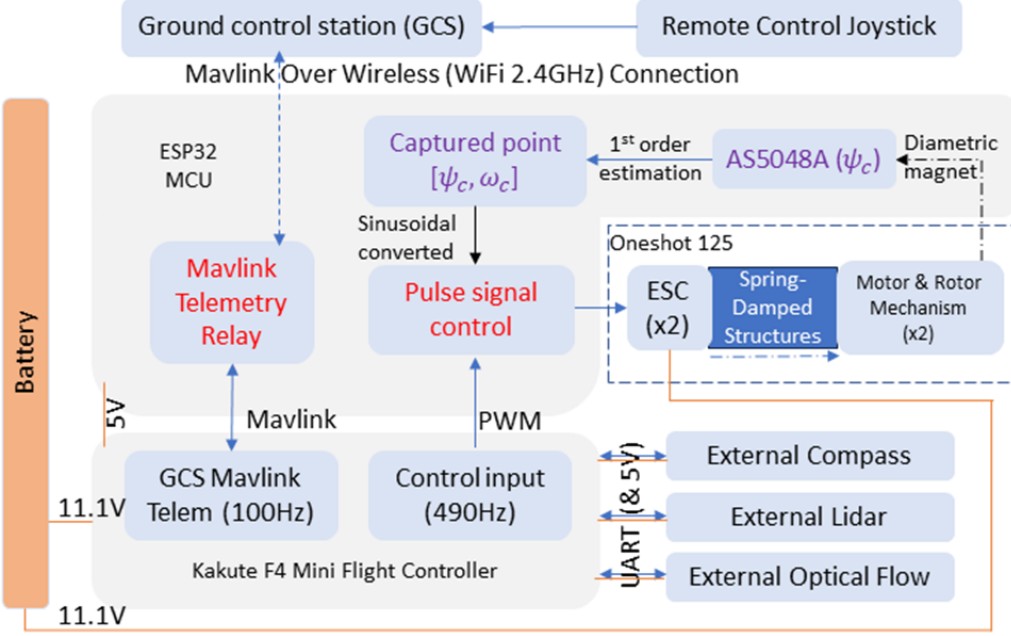

**Figure 3.** The bicopter platform's avionics.

## 3. Materials and Methods

This study uses a unique underactuated swashplateless rotor mechanism with a generic 8 × 4.5 inch multicopter propeller, as shown in Figure 1, on a tiltrotor bicopter platform. Unlike the conventional underactuated swashplateless rotor mechanism, this unique constrained 3DoF sphere joint mechanism is designed to maintain a consistent pitch angle on each blade and maneuver the platform with vectored thrust, as shown in Figure 4 (left).

### 3.1. Constrained 3DoF Underactuated Swashplateless Propeller Mechanism

Figure 4 (right) shows a zoomed-in view of the rotor hub mechanism with rolling contact surfaces with ball bearings. The upper part of the rotor hub mechanism with a contact surface is machined out of aluminium 7075. Unlike conventional hinged designs, the hub operational mechanical damping coefficient can be reduced to consider small

values. During rotor tilting angle ground tests, no significant friction bound is observed while tilting amplifying any specific rotor tilting angle [10]. In the simulation model, zero amplitude is also meaningful (thrust vector centred), the control signal sent to the ESC can be set continuously, and the simulation model still correctly predicts the rotor behaviour using the raw control mapping from the flight controller.

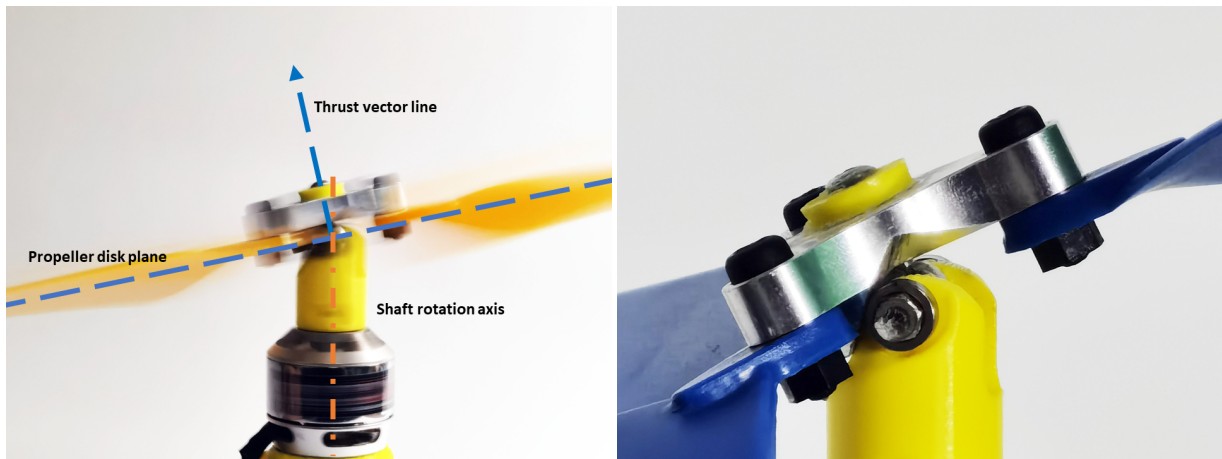

**Figure 4.** Constrained 3DoF rotation vectoring mechanism. (**left**) Demonstration of stable thrust vectoring without cyclic blade pitch. (**right**) A close view of constrained 3DoF mechanism used in this study.

*3.2. Simulink Simscape Modelling*

In conventional underactuated swashplateless rotor designs, an orthogonal lead–lag axis motion results in linear changes in the blade angle of attack while considering small deflection angles. Cyclic aerodynamic lifting force on each blade is included in the model [10]. However, multiple-blade propellers maintaining a static thrust vector are also considered in this model. In contrast, blade pitch angles remain steady; the same assumption can only correctly apply to the blade at a 0 or 180 degree phase angle and only holds for some of the blades. However, in this underactuated propeller, all blade angles in 3D vectors can always be treated as directly rotated along one single vector located at the spherical centre of the mechanism joint from the inertial earth frame of reference. Therefore, in the simulation model of this study, it is straightforward to treat the transformation directly as a rotation manipulation. A MATLAB Simulink Simscape model is constructed, as shown in Figure 5.

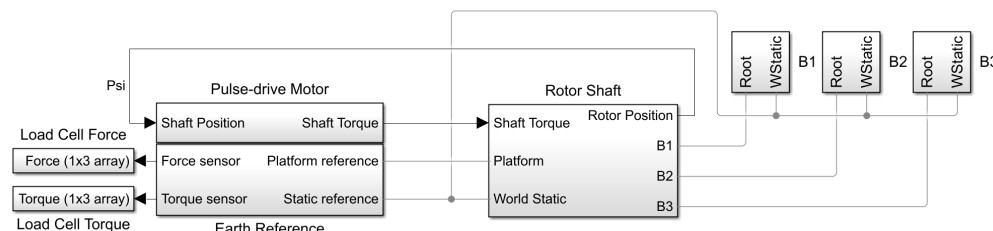

**Figure 5.** Simulink Simscape block diagram overview

The rotor model is built based on the MATLAB Simulink (Software version R2021b) modelled blade element method (BEM) with blade elemental momentum theory (BEMT) implemented on each separate blade and integrated to represent a single rotor of the platform [14–17]. Each blade diagram (B1, B2, and B3) is identical but attached to the rotor hub according to its CW/CCW spin direction. Figure 6 shows multiple sub-block of elements of blade B1 connected in series representing the whole blade aerodynamic load, and Figure 7 shows the internal components of the blade element block.

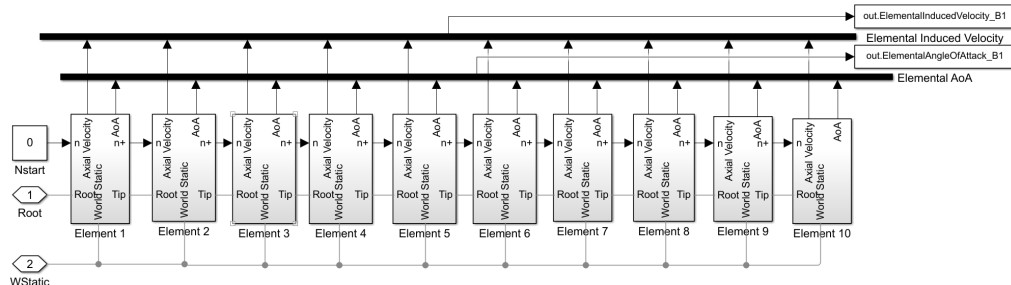

**Figure 6.** The Simulink Simscape block diagram with 10 aerodynamic blade elements connected in series representing a single blade aerodynamic model (Inside "B1" block of Figure 5).

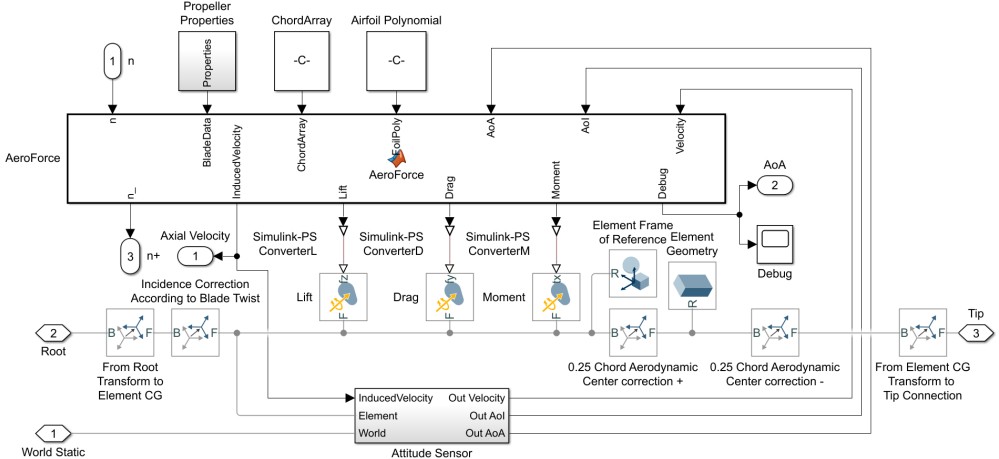

**Figure 7.** A Simulink Simscape blade element (Inside "Element 1" block of Figure 6).

The aerodynamic forces on each blade use the elemental aerodynamic equation:

$$\vec{F} = \frac{1}{2}\rho \vec{V_e}^2 s_e C_e F_{tip} F_{hub} \tag{1}$$

where $\vec{F}$ represents the three-dimensional aerodynamic load $[F_y, F_x, M_z]$, which is usually considered $[L, D, M]$ for lift, drag, and moment load, respectively. Air density $\rho = 1.225$ kg/m$^3$ is assumed constant in the simulation. $s_e$ represents local element area, $C_e$ is composed of three terms for three dimensions representing the elemental local lift coefficient $C_L$, drag coefficient $C_D$, and chord $c$-scaled moment coefficient $C_M \times c$. All coefficients are linearly interpolated based on the local Reynolds number $Re$ and the local true angle of attack $\alpha$ from the AIAC airfoil database of the airfoil "ag11". $Re = \frac{Vc}{\mu}$ where $\mu = 1.4607 \times 10^{-5}$ m$^2$/s is considered the constant flow viscosity at sea level. Both $F_{tip}$ and $F_{hub}$ are Prandtl's tip loss factor [18], applied based on the elemental distance to the blade edge as in Equation (2).

$$F_{tip} = \frac{2}{pi}cos^{-1}(e^{\frac{-B(R_{tip}-r)}{2r|sin(\alpha)|}}), F_{hub} = \frac{2}{pi}cos^{-1}(e^{\frac{-B(r-R_{hub})}{2r|sin(\alpha)|}}) \tag{2}$$

where $B$ represents the rotor blade number, and $R_{tip}$ and $R_{hub}$ represent the radius of the blade tip and hub, respectively. $\alpha$ is the element's real angle of attack. From Equation (1), airfoil lift and drag forces can be converted to geometric lift and drag force, directly applied to align to the airfoil chord line, rotated by the angle of attack $\alpha$. Since the airfoil moment applies to the same axis of conversion, it remains unchanged after the rotation along the blade axis:

$$L_G = F_y * cos(\alpha) + F_x * sin(\alpha) \tag{3}$$

$$D_G = F_x * cos(\alpha) - F_y * sin(\alpha) \tag{4}$$

$$M_G = M_z \tag{5}$$

To work out an estimation for the local induced velocity, rotational direction element lift (or axial direction element lift) is calculated to evaluate the extent to which induced flow speed per element contributes to the total:

$$L_e = L_G * cos(\alpha_I) - L_G * sin(\alpha_I) \tag{6}$$

where $L_e$ stands for the axial direction element lift, and $\alpha_I$ represents the element pitch angle, which can be considered a fixed angle, as the propeller blade does not have an additional degree of freedom (DoF) in teetering motion. Therefore, no rotation frame of reference transfer is needed to convert from an element-axial vector to a rotor-axial vector. From the elemental disk theory and equations above, induced velocity can be estimated as follows:

$$V_I = \sqrt{\frac{BL_e}{2\pi\rho r l}} \tag{7}$$

The frame of reference transform model using the cross product of the elemental velocity vector in the inertial 3D space corrected by elemental pitch angle is implemented to detect the blade element 3D linear velocity and the 3D angle of attack to generate aerodynamic forces, as shown in Figure 8. A final rotor model is established with BEM with ten small sections of the blade, each having a drone propeller airfoil sized to a 1/10 section of the actual blade, assuming the use of an ag11 airfoil, which has a similar maximum thickness (5.8% at 26% chord) and chamber (2.2% at 29.7% chord) to the actual physical blade. Aerodynamic loads on each element are corrected with Prandtl's tip-loss factor, as in Equation (1).

The ground reference of the system is demonstrated in Appendix A Figure A3.

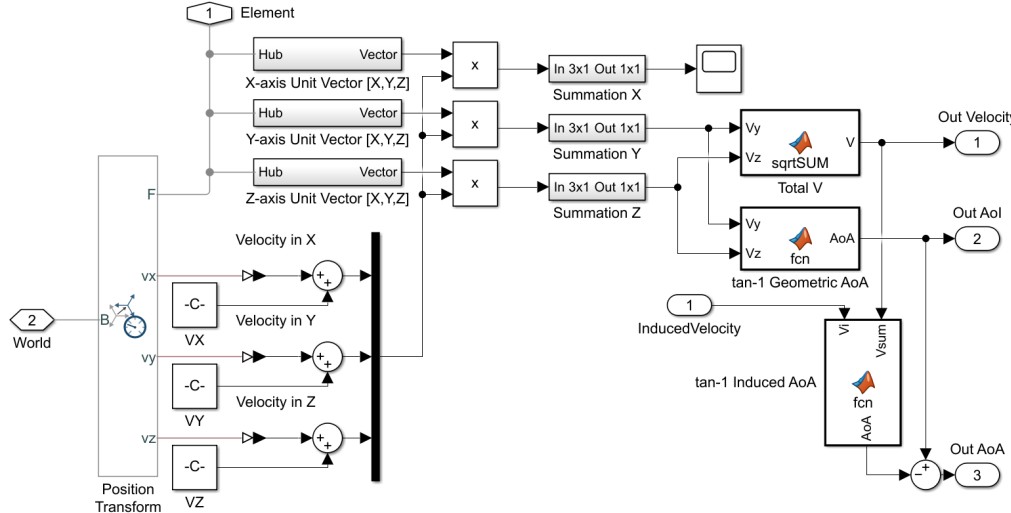

**Figure 8.** An elemental angle of attack (AoA) detector in Simulink Simscape (Inside "Attitude Sensor" block of Figure 7).

### 3.2.1. Model of Spring-Damping Structures in Simulation

To study the effect on the system when a spring-damping system is included, a passive degree of freedom with constant spring and damping coefficients is modelled beneath the Simulink Simscape main rotor shaft, as shown in Figure 9. Methodologically, there are two ways in which the spring-damping system can be assembled. When the motor-magnetic encoder (angular rotation sensor) is rigidly attached, the magnetic encoder can only receive the position differences between the rotor shaft and the damper attached along the rotor shaft, which is shaft to damper position $\psi_s$. However, to occupy the inertial angle of the rotor shaft in the platform frame of reference, the critical sensor data are the shaft position and velocity reading in the platform inertial frame of reference, which provide both the

shaft-to-damper position sensed from motor shaft $\psi_s$ and the damper position sensed from the spring-damping mechanism $\psi_d$, as shown in Figure 9.

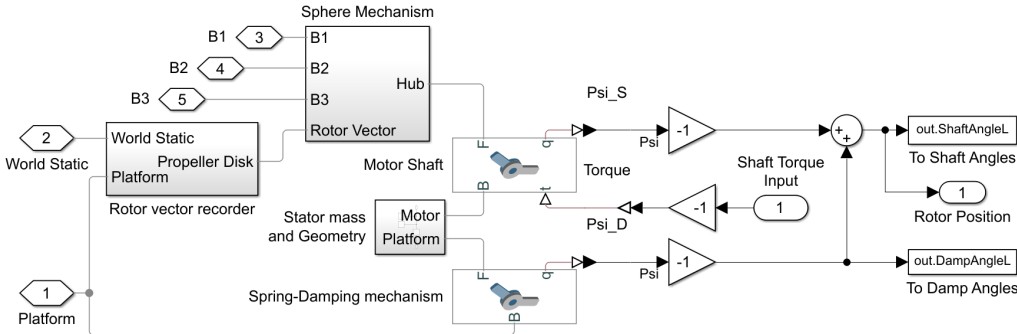

**Figure 9.** The Simulink Simscape spring-damped rotor shaft ("Rotor Shaft" block of Figure 5).

### 3.2.2. Model Swashplateless Mechanism in Simulation

To correctly simulate the response of the rotor hub mechanism, the "Sphere Mechanism" block, as shown in Figure 10, connects the rotor shaft frame of reference with the rotor blade frames of reference. On constrained 3DoF rotor hubs, the spring coefficient of the 3DoF spherical joint is set to 0. The damping coefficient of the 3DoF spherical joint needs to be obtained through experiments. Therefore, an experiment is designed to match the damping coefficient between the simulation model and the physical rotor. Under solid stator damping conditions, at an average rotor speed of 70 Hz, the rotor speed is amplified to a range of vectoring angles in degrees. In contrast, the actual shaft speed amplitude under each vectoring angle is recorded. In the simulation, under the same condition and rotor model, another map of the rotor vector tilting angle can be obtained by varying the rotor underactuated mechanism damping coefficient and pulse-driven shaft speed amplitude. By matching the experimental rotor tilting angle and simulated rotor tilting angle, an effective swashplateless mechanism damping coefficient can be obtained.

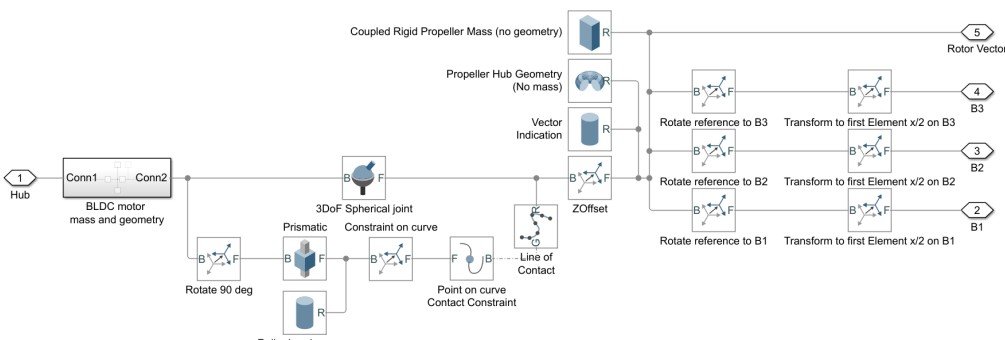

**Figure 10.** The Simulink Simscape model of the constrained 3DoF rotor (Inside "Sphere Mechanism" of Figure 9).

### 3.3. Components Design and Integration

A motor mount is constructed to experimentally validate the effectiveness of spring-damping structures, as shown in Figure 11. The CAD model in Figure 12 shows the internal structure of the passive damping structures underneath the motor. An elastic damping material is placed in each slot, as shown in the CAD model. The spring coefficient can be trimmed by changing or resizing the damping material filled inside each slot. By tightening up the motor mount, this structure can be firmly attached and, thereby, considered solid while keeping the angular frame of reference of the controller unchanged to make comparisons easier.

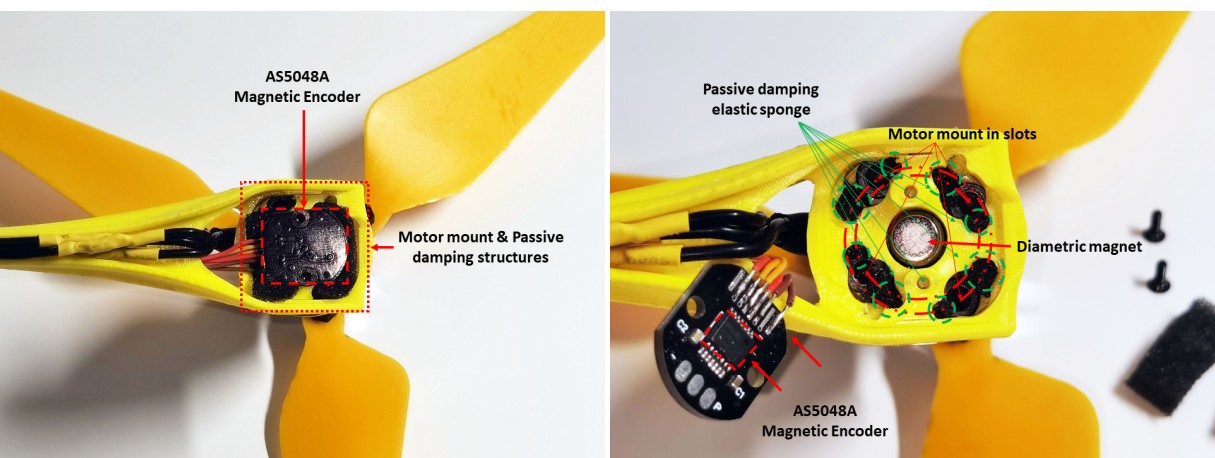

**Figure 11.** Motor mount structures with components installed. (**left**) Bottom view of the platform propeller axis with magnetic encoder installed. (**right**) Exposed view of the internal strucutres with damping materials filled in.

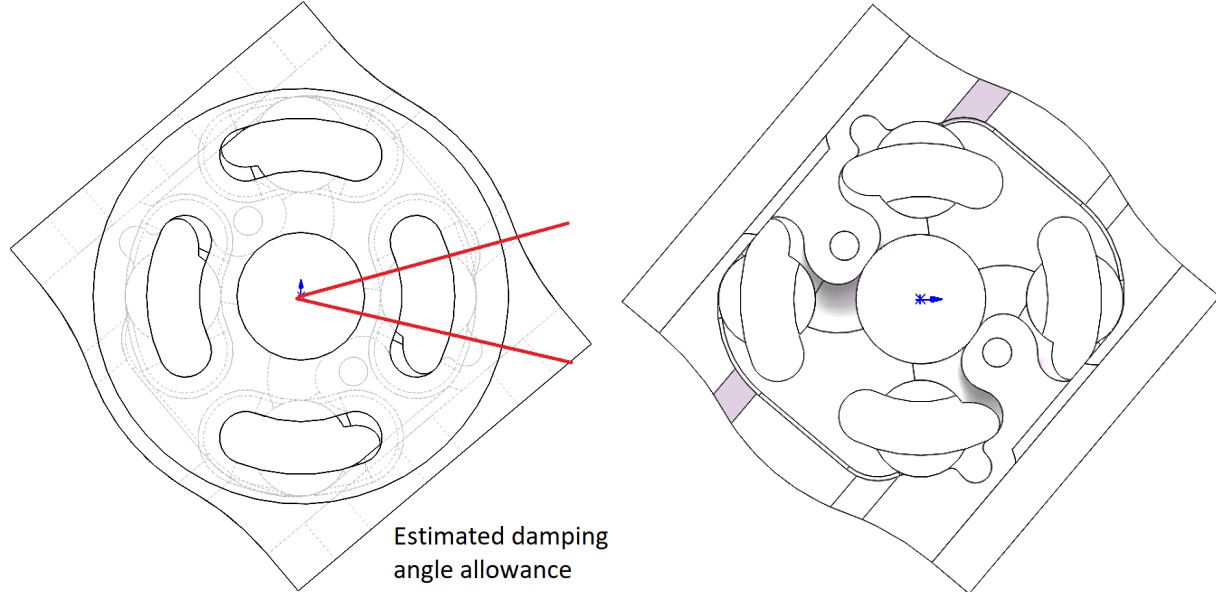

**Figure 12.** Computer-aided-design (CAD) drawing with different view angles showing extra modelling details. (**left**) Top view in Solidworks CAD software. (**right**) Bottom view in Solidworks CAD software.

### 3.4. Simulation and Experimental Setup

This study consists of two major parts. Between the simulation and physical model, we run both models to the same propeller speed while actively tilting both mechanisms to the same angle (shaft speed amplitude, phase, and bias). In the MATLAB Simulink section, to study the effectiveness of using a spring-damped constrained 3DoF linkage as an underactuated mechanism at around 70 Hz (4200 RPM) rotor speed tilted to 12 degrees, different spring-damping coefficients are studied while its external load is recorded, as shown in Figure 13.

From Simulink data, a vibration spectrum can be derived where the first three spectra can be obtained, as shown in Figure 14. The contour plot region represents the spectrum gains of the vibration. In return, reducing the vibration's first or second peak of spectrum gain can simplify the load condition. However, controlling the first peak with the highest amplifying gain is usually the most desired outcome.

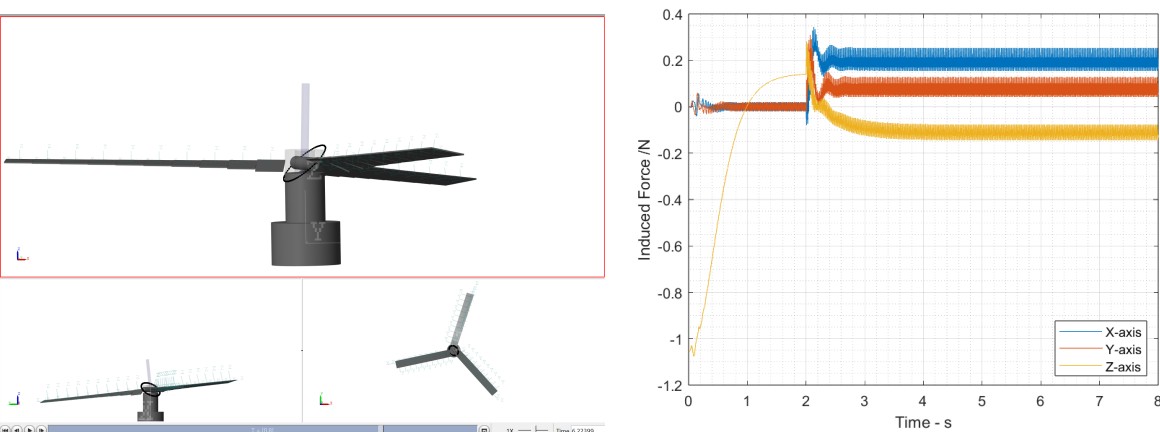

**Figure 13.** A force scope example of simulation output. (**left**) Simulation mechanics explorer. (**right**) Simulation force-scope output.

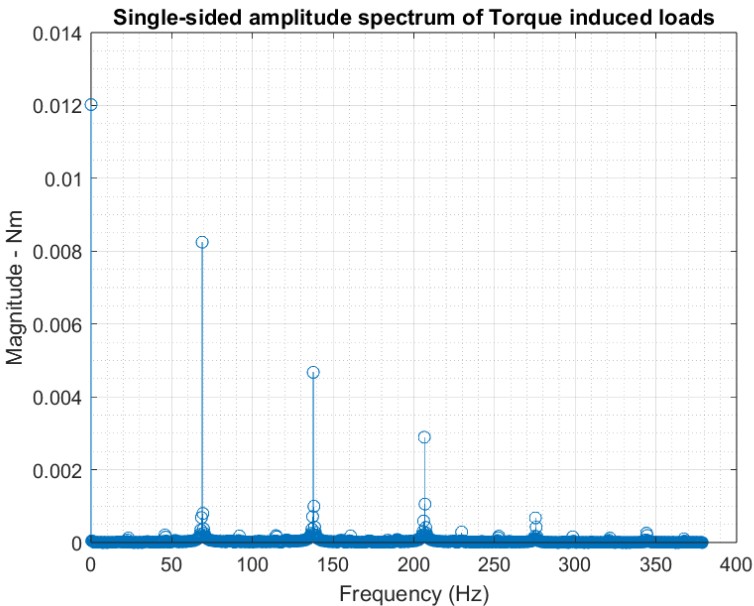

**Figure 14.** A single-sided spectrum plot example of total torque simulation output.

Upon observing the experimental results, the axial spring coefficient can be measured after securing the fill-in material. However, estimating the actual friction damping coefficient is more complex, since it depends on both the component wear and tightness. This value also slightly varies over time. Therefore, test results are collected by conducting multiple tests when all result-matching-estimated friction-damping coefficients are similar and no longer change. Thereby, by matching the angle of the stator motion during the amplification, the stator damping coefficient can be estimated.

In this study, the absolute vibration can be used to evaluate real vibration magnitude, since the study controls all internal variables except spring-damping mechanism properties. The vibration magnitude is estimated using measured data from on-board IMU [19] in units of m/s$^2$. Analysis with a Batch sampler provides vibration FFT results [20] that can be used for amplification comparisons described in units of decibels for each power level [21]. Generally, according to the flight controller source page, we use absolute magnitude levels 15 and 30 m/s$^2$ as the baseline values for high-quality and low-quality IMU noise levels, respectively. As for the noise level described in decibels, the valid threshold for an acceptable IMU signal quality is at a 0 dB power level. The single spectrum power at 0 dB indicates the same measured quantity and reference quantity [22]. The vibration is neutrally

well-filtered over a valid series of IMU data, whereas the filtered signal above 0 dB cannot be considered stable. Below the critical vibration power level, the lower peak indicates better damping filter quality; the quantified relative comparisons are made to determine the effectiveness of the load alleviation by installing the spring-damping structures [23].

## 4. Results

The prototype platform used in this study in Figure 1 has a total weight of 248.4 g, with 3 to 5 g spring-damping structures in total (varies depending on fill-in material) and static hovering at 4200 rpm. The rotor diameter is 24 cm, powered by a 3 cell 460 mAh battery; it provides a maximum thrust of 4.14 N (maximum thrust-to-weight ratio of 1.7). The swashplate mechanism used in this study is assigned to rotate at 70 Hz, a 12 degree banking angle. Ground tests with various operating angles are captured to demonstrate the operation of a spring-damped underactuated swashplateless rotor. The slow-motion video link is provided in Supplementary Data of this paper in the Supplemental Materials Section. As for ground test results, phase angle induces no significant impact on rotor tilting angle against torque modulation.

### 4.1. Rotor Mechanism Damping Coefficient Estimation

As shown in Figure 15, under different damping coefficient conditions, the rotor vectoring angle and the rotor shaft speed cyclic component at a stable 70 Hz rotor speed formed gradient correspondence. Ten separate ground tests are carried out and overlayed on the contour plot. In 10 separate ground tests of the exact mechanism at a 70 Hz average rotor speed at multiple resultant rotor tilting angles, the rotor speed is collected through a magnetic encoder, as shown in Figure 16, and the rotor angle is estimated optically, similar to Figure 4. The collected results are overlapped on the contour plot in Figure 15. Both simulation and static ground tests have fully secured motor mounts (solid).

As for the results, the rotor underactuated mechanism is tested to have an average damping coefficient of $8.6 \times 10^{-7} \frac{\text{Nm}}{\text{deg/s}}$ on a lubricated rolling contact. This value is the rotor hub mechanism damping coefficient for all simulations exploring the impact of stator spring-damping setups.

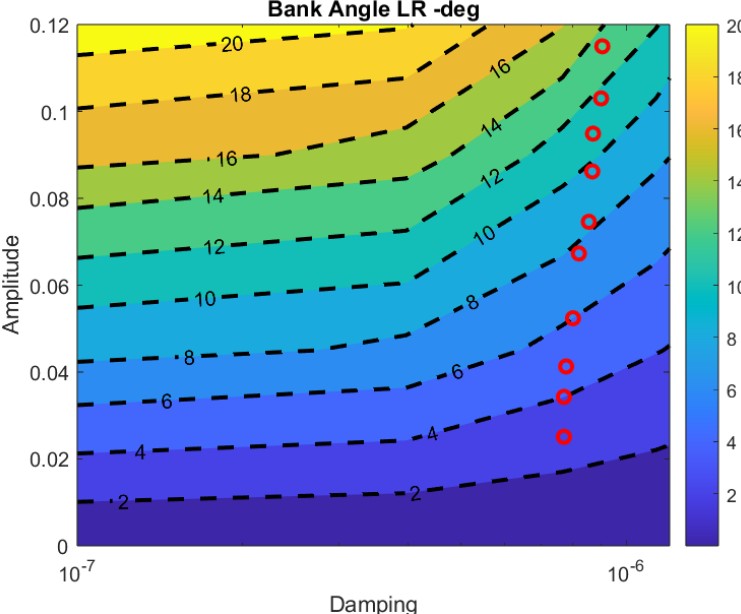

**Figure 15.** Bank Angle contour plot with cyclic rotor speed amplitude (at 70 Hz) against Damping coefficients in simulation. Red circles in figure represents each ground test points collecting cyclic rotor speed amplitude (Amplitude) and Rotor Bank Angle.

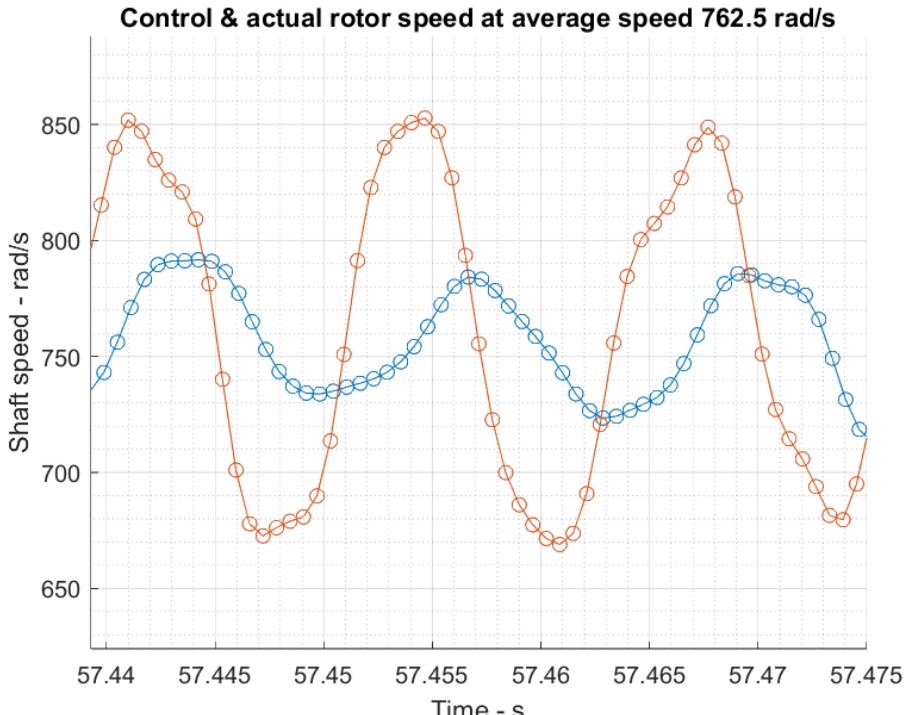

**Figure 16.** An example of magnetic encoder-collected actual rotor speed at a single test point when ESC is given a target rotor speed with the rotor tilting angle measured to be 9.6 degrees. (Red: ESC control speed. Blue: encoder collected real shaft speed).

*4.2. Simulation Results*

As shown in Figure 17, the contour plot shows a cyclic torque load peak amplitude rise between the stator and platform crossing stator damping coefficient from $1.0 \times 10^{-7}$ to $5.0 \times 10^{-4} \frac{\text{Nm}}{\text{deg/s}}$. Figure 18 shows a contour plot with the angle output of the spring-damping mechanism, and Figure 19 shows its actual form. This result shows that significant angular motions are suppressed at a higher stator damping coefficient. Naturally, this would result in more cyclic and vibration-inducing loads being transferred into platform structures.

As shown in Appendix A Figure A2, the force-induced vectoring vibration does not vary significantly under different stator damping and stiffness coefficients. Although the total spectrum magnitude can still be captured, the variation remains mainly within 0.067 N to 0.07 N, vibrating within a range of about 6.8 g (over a total of 120.4 g) of generated thrust. In Appendix A Figure A1, it is straightforward to see that changing the spring-damping characteristics of the structure has little impact on the rotor tilting capabilities. The simulation results show that vibration is transferred at high levels to the platform after the stator damping coefficient (axial friction coefficient) rises above a certain level. Then, using the estimated friction damping coefficient, the model can be validated from this study regarding stator stiffness. The effectiveness of the spring-damping structures can be evaluated by comparing the IMU-collected vibration spectrum with various stator stiffness levels.

In both Figures 17 and 18, from simulation, a highlighted high-value region is revealed between damping coefficients of $1.0 \times 10^{-7}$ and $8.0 \times 10^{-6} \frac{\text{Nm}}{\text{deg/s}}$ and between spring coefficients of $1.5 \times 10^{-3}$ and $4.5 \times 10^{-3} \frac{\text{Nm}}{\text{deg}}$. This indicates that the spring coefficient at around $3.0 \times 10^{-3}$ is the system's natural frequency, taking actual rotor parameters into account. Therefore, although the lowest vibration region of the result occurs at the lowest damping coefficient regions on the actual rotors, the spring coefficient must also be sure to be below $1.0 \times 10^{-3}$ to avoid resonance.

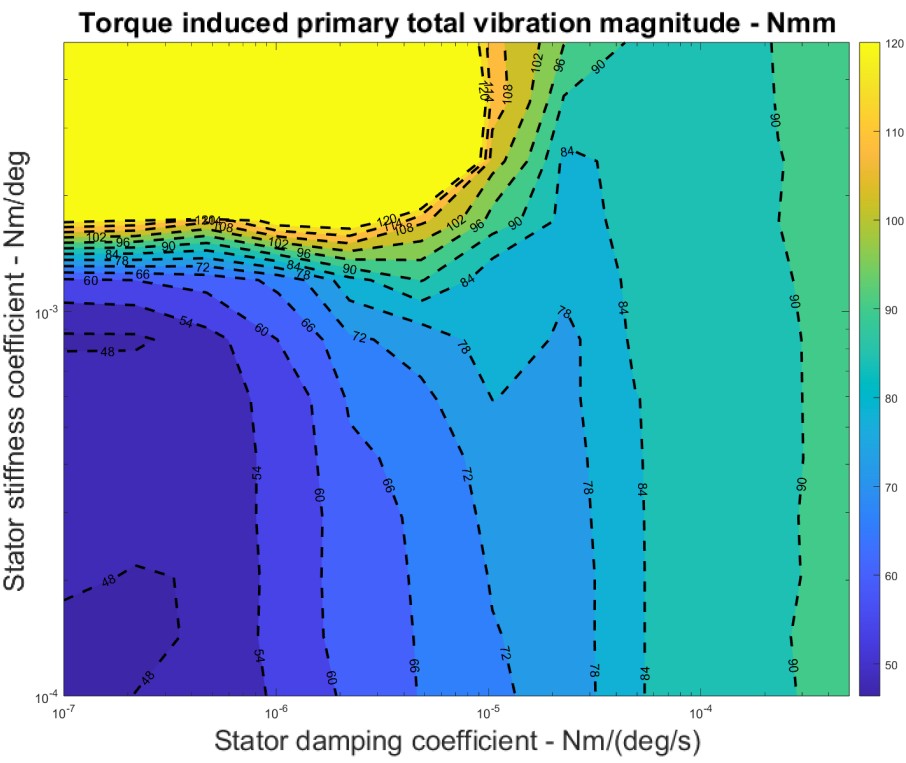

**Figure 17.** The magnitude of the vibrating torque load at the main peak frequency that is transferred into the main structure (the figure is saturated above high magnitudes, as the values are unrealistic in representing a linearized model).

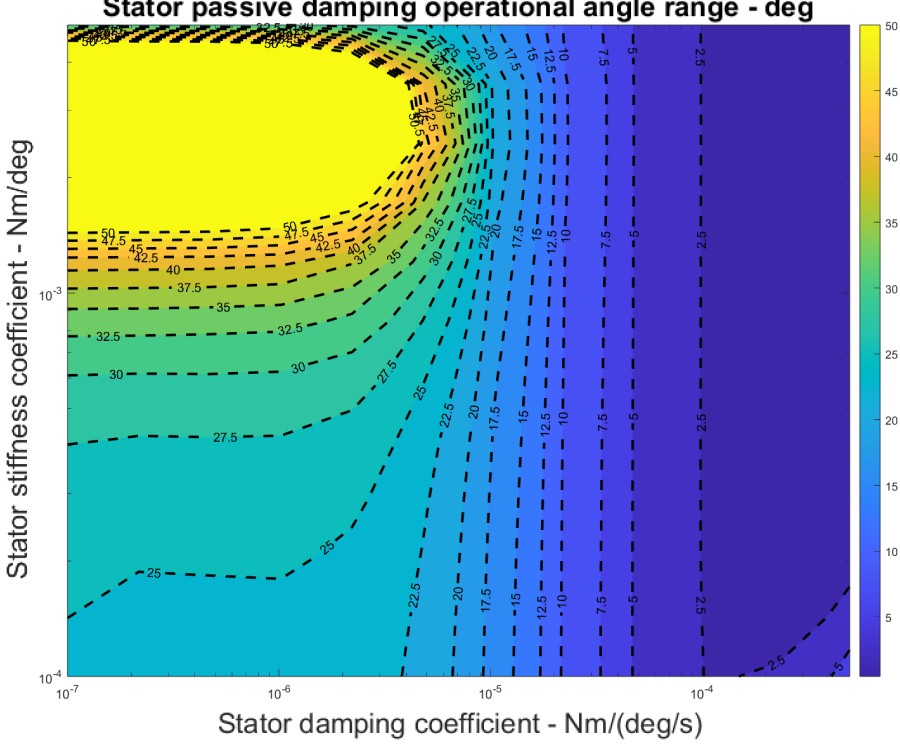

**Figure 18.** Stator cyclic motion range under current profile (the figure is saturated above high angles, as the values are unrealistic in representing a linearized model).

It is also essential to ensure that the stator angles are within range while choosing the designs of the spring-damping structure. In Figure 18, it is clearly shown that reducing

the vibration magnitude to minimal ranges has a significant cost of increasing stator cyclic responses to similar rotational motions, as shown in Figure 19. Therefore, a highly effective spring-damping structure is physically challenging due to effective coefficient control and damping angle range constraints.

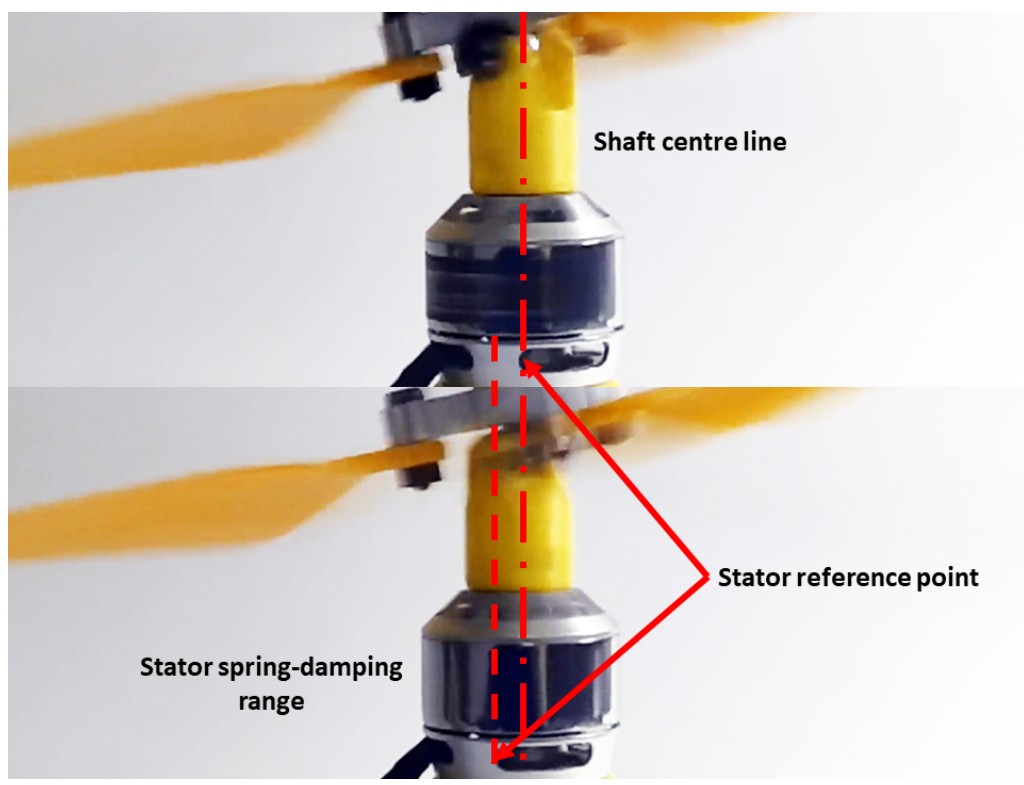

**Figure 19.** An example showing the angular motion of the spring-damping structures.

Using the corresponding test parameters, validation and observational data are collected through onboard IMU on-ground and hover tests. (Supplementary Materials: Videos S1–S3).

Stator Damping Coefficient Estimation

Figure 19 is an example of one of the four separate tests, given a measured axial spring stiffness of $7.9 \times 10^{-4} \frac{\text{Nm}}{\text{deg}}$, operating under the prescribed conditions. Photos show an oscillating stator angle range between 12 and 13 degrees. Overlapping this result with simulation results gives Figure 20 (left). This indicates that the stator friction-damping coefficient is between $1.8 \times 10^{-5}$ and $2.2 \times 10^{-5}$.

By tightening the mounting structures completely, according to the simulation results in Figure 17, the vibration magnitude should increase from around 77 Nmm to above 92 Nmm, indicating an increase by more than 19.4% of the total vibration magnitude. In reverse, by adding the spring-damping structures, the expected vibration magnitude reduction in total is at least 16.3%.

Single-rotor results are obtained from ground tests. With the platform placed on the ground, the rotor is instructed to execute a steady profile with an average rotor speed of 70 Hz and a tilting angle of 10 degrees. This section consists of a series of test setups and the resultant 60 s average IMU-collected vibration peak magnitudes listed in Table 1:

**Table 1.** Single rotor experimental setup.

| Variables | Set 1 | Set 2 | Set 3 | Set 4 | Unit |
|---|---|---|---|---|---|
| Stiffness | Tightened | $1.7 \times 10^{-3}$ | $9.4 \times 10^{-4}$ | $7.9 \times 10^{-4}$ | $\frac{Nm}{deg}$ |
| Damping | Tightened | $2.0 \times 10^{-5}$ | $2.0 \times 10^{-5}$ | $2.0 \times 10^{-5}$ | $\frac{Nm}{deg/s}$ |
| Mean IMU acceleration magnitudes | 19.5 | 12.81 | 11.10 | 10.79 | $\frac{m}{s^2}$ |

Figure 20 (right) shows the corresponding comparison range with markings indicating where the IMU acceleration magnitudes are collected.

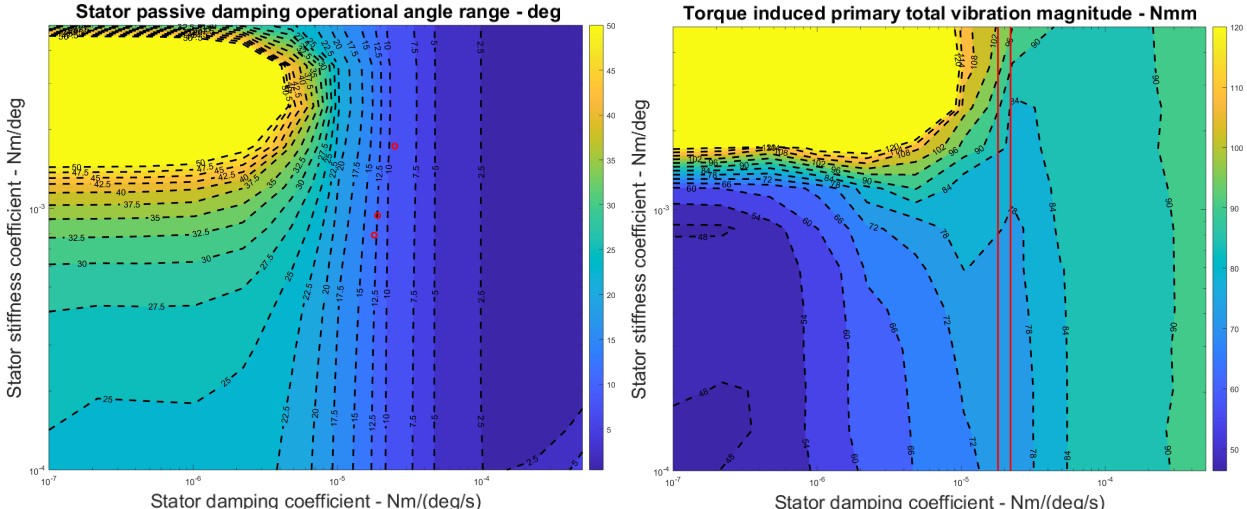

**Figure 20.** Vibration magnitude collected during on-ground tests overlapped with simulation results. (**left**) Test results of stator damping angles overlapped with simulation results to obtain stator friction-damping estimations. (**right**) Expected data range (comparison group) overlapped with stator damping coefficients around $2.0 \times 10^{-3} \frac{Nm}{deg/s}$.

From the vibration magnitudes collected from IMU in the form of peak local accelerations, we can observe that, compared with simulations, the reduction in peak acceleration magnitudes follows the indication by contour plot results. This result also indicates that although physically limited by the damping angle allowance (currently about 15 degrees—given in Figure 12 (left)), reducing the spring coefficient still successfully reduces the load transferred to the platform. However, if a higher damping angle is possible, it may further reduce the cyclic load and improve the vibration filtering capability.

*4.3. Platform Test*

During platform operation for vibration tests, a comparison is made based on the operational vibration reading of the same platform with the motor mount damped at $7.9 \times 10^{-4} \frac{Nm}{deg/s}$ and tightened, respectively. During both tests with Ardupilot-controlled hover, the platform takes off from the ground in QHover mode and maintains an altitude of around one meter high. As shown in Figures 21 and 22, the vibration recorded by onboard IMU, represented in the flight controller unit acceleration noise form, differs based on how the stator is installed. The peak vibration magnitude collected when the motor stator is passively spring-damped is notably reduced by 28.8%. Moreover, the peak vibration axis on the flight controller now shifts from the X axis to the Y axis. In both figures, in the platform hovering frame of reference, X represents its upward direction, and Y and Z represent the lateral sideways and forward directions, respectively. The former peak vibration axis occurs on the X axis with an average of $24(\frac{m}{s^2})$. After spring-damping structures are installed, this

value is reduced to an average level of $13(\frac{m}{s^2})$. Comparing the vibrations in Y and Z, the total average magnitude is not significantly impacted, still maintaining at about $15(\frac{m}{s^2})$ and $5(\frac{m}{s^2})$, respectively. This indicates that installing the spring-damping structures effectively reduces the noise induced parallel to the motor axis. Knowing the operating principle, it is reasonable to speculate that the spring-damping structures result in the reduction in motor axis vibration power. Moreover, the acceleration reading width of each axial vibration signal recorded, representing reverberation intensity between axes, is also significantly reduced. This indicated that the control-inherent vibrations and induced noise are reduced. As a result, since the spring-damping structure kinematically separates each source of vibration axis over the main structure (platform fuselage), this structure also reduces the reverberation while multiple motors operate simultaneously. By observing the FFT results as shown in Figures A4–A7, the first peak magnitudes undertaken by IMU in acceleration X–Y–Z axes and gyroscopic X–Y–Z axes are both reduced by 10 Hz at the consistent hovering peak frequency (reduced from 0 dB to −10 dB in ACC measurements, reduced from −30 dB to −40 dB in GYR measurements). The second peak magnitude collected above the main peak frequency (rotor frequency) is not significantly impacted (both peak at −30 dB for ACC measurements and −50 dB for GYR measurements). Therefore, from the 10 decibel peak power reduction, we can expect the signal vibration to be "half as loud" in its main peak induced in the same frequency. Matching the peak frequencies collected during flight tests and simulation also implies that the multiple peak simulation spectrum of the rotor can be used to estimate the absolute vibration peak frequency and magnitudes after calibrating with readings from the actual rotor. The results generally indicated that the vibration load is alleviated after the spring-damping structures are installed.

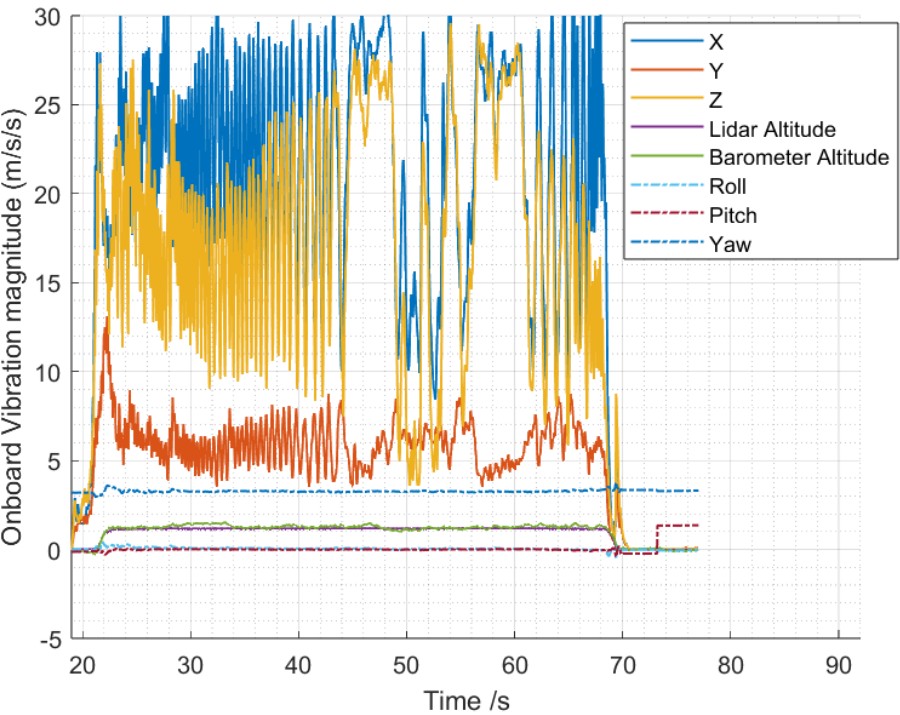

**Figure 21.** Flight controller collected in-flight vibration data VIBE magnitude over time of tightly-fixed motors.

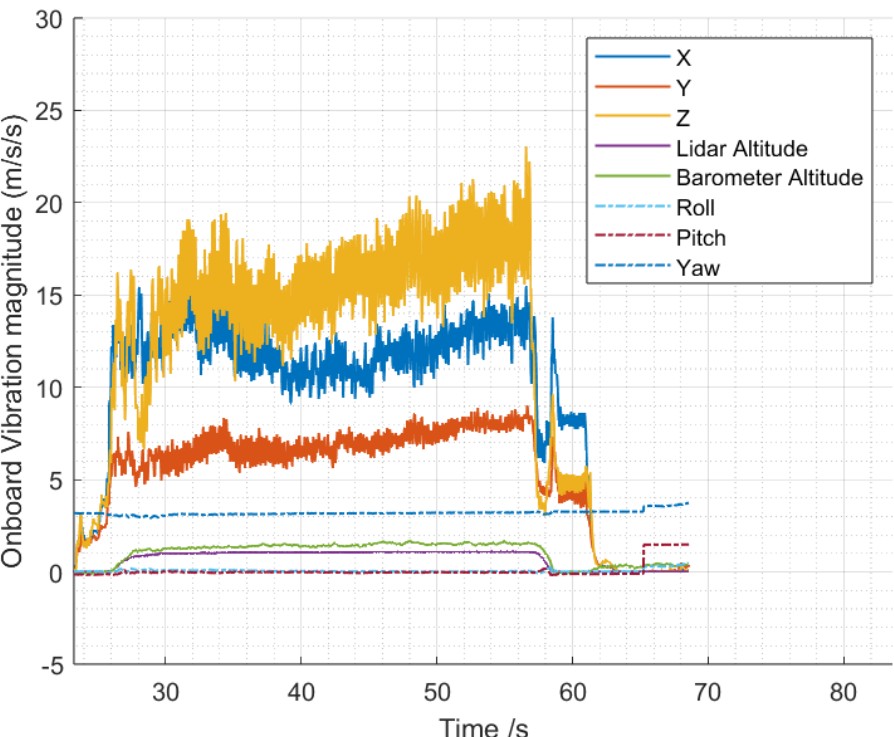

**Figure 22.** Flight controller collected in-flight vibration data VIBE magnitude over time of spring-damped motors.

## 5. Discussion and Conclusions

Comparing the experimental results of motor angular velocity and vibration magnitude with various spring-damping coefficients in the shaft axis, the results provided fundamental theories for the system design of vibration isolated from the underactuated thrust vectoring rotor to minimize the vibration received by onboard avionics.

- The simulation in this study provides the results of the parameter study. The simulation shows that the most effective spring-damping structure that best fulfils the control-inherent vibration reduction purpose should have smooth damping and an appropriate spring coefficient magnitude to avoid resonance and make the structural design affordable, appropriate, and acceptable. The design region (range of coefficients) is characterised through simulations with all the parameters validated by experiments. This resultant design region is challenging to fully explore with only experiments or simplified linearized models, as simulations can provide numeric estimation based on finite element methods. Comparing the idealized single-sided spectrum output of the simulation in Figure 14 with Appendix A Figures A4–A7, it can be shown that simulation results successfully predicted the multiple frequency spectrum peaks instead of only the main peak.

- The simulation in this study shows that even with ideally continuous and clean input torque as simulated, without any internal or external background vibrations, the minimum control-coherent vibration magnitude is still inherent and inevitable, as shown in Figure 14. Because the shaft torque modulation is the source of both system control and the control-inherent vibration at the same time, any study focusing on vibration issues in torque-modulation-type underactuated swashplateless rotors and propellers should separately understand the impact of the control-inherent vibration and motor-propeller natural vibration.

- One of the main contributions of the experiment in this study is in determining the friction-damping coefficient in the simulations. It is worth noting that the purpose of back-calculating (estimating) the friction-damping coefficient is to find the value that fits the physical model best at a design point. On one hand, this value might vary

over time, requiring monitoring and trimming over different tests or physical model variations. On the other hand, this value is only of a limited reference significance instead of being an accurate, scientifically measured friction-damping coefficient. The complexity of friction-damping physics is a barrier in value estimation. Luckily, from the parameter study results, a conclusion can be made that under most design conditions, the optimal friction-damping coefficient is always "as low as possible".

- Most importantly, a different case arises when changing the spring coefficient to avoid resonance is not feasible. In this case, the friction-damping coefficient must be high enough according to Figures 17 and 18. Otherwise, the rotor may be subject to severe damage or even be at risk of destruction. In the same way, for an aircraft with a prescribed mission profile (maximum and minimum rotor frequency, etc.), the simulation must be thoroughly used to determine whether an operational status puts vehicle components under dangerous conditions.

- The experiment in this study validated the hypothesis that adding a spring-damping structure in its torque modulation axis (central shaft axis) can absorb the stator's excessive cyclic lead–lag motion before transferring the load into the platform's main structure.

- The design parameter study shows that the control-induced vibration of the prototype can be reduced at hovering frequency. The prototype in this study achieved 10 decibels of peak power reduction comparing two identical free-body hover flight missions maintained at around 1 m high altitude. Thereby, although physical angular constraints still limit the current spring-damping structure design, the main structure-vibrating motion is already reduced to an acceptable level, taking the guide manual of the current Ardupilot firmware as a reference. This also indicated that further optimisation can be carried out if conditions permit.

- Given that the minimal proportion of spring-damping structures in this study is 3–5 g over 240.8 g in total weight, the weight proportion of spring-damping structures can be considered insignificant. However, if a larger passive stator damping angle is considered, the structural weight of this component might drastically increase depending on whether it is cleverly designed; thereby, the actual platform performance might be subject to changes.

These results imply that self-coherent vibration can be alleviated with a particular spring-damping design. It is worth noting that this process also changes the signal obtained by sensors. Therefore, an underlying control variable calibration is required after the spring-damping structures are installed or uninstalled. After a change in motor or rotor mass inertia properties or a change in the modulation algorithm, the system frequency response must also be re-estimated to keep the design point away from the resonance region.

## 6. Future Developments

Based on the research tool developed in this project, other mechanism variants could also be invented to pursue other operational characteristics. This includes, but is not limited to, developing a feedback thrust vectoring controller in high-velocity incoming flow or side flow, thrust controlling and damping optimisation in forward flight, modifying the 3DoF constraint condition curvature, or applying spring stiffness to the 3DoF constraint condition to generate a higher thrust-vectoring angle. However, the full development of these applications requires a new methodology to obtain a full-state (feedback) estimation of the propeller vector.

According to the scalability study of a conventional rotor design proposed in another research paper, aircrafts with larger weight and slower rotors place a proportionately smaller torque modulation load on the drive motor than smaller, faster rotors. The dynamics are fundamentally insensitive to the gross scale of the aircraft. Therefore, this technology might be suitable for tiny and huge aircraft [4]. In the fundamental theory, both the conventional design proposed by James Paulos in 2013 and the innovative constrained 3DoF rotor design proposed in this paper require precisely the same waveform and proportional

load increments as the tilting angle increases. Thereby, as a future direction, the constrained-3DoF joint technology can be implemented on passenger-sized tiltrotor eVToL vehicles if a sufficiently high proportional range can be experimentally explored and affirmed.

**Supplementary Materials:** The following supporting information can be downloaded at: https://www.mdpi.com/article/10.3390/machines12050296/s1, Video S1: Example slow-motion video of operating spring-damped underactuated swashplateless rotor, modulated at 90-degrees phase angle, 15-degrees tilt angle. Video S2: Example slow-motion video of operating spring-damped underactuated swashplateless rotor, modulated at 135-degrees phase angle, 15-degrees tilt angle. Video S3: Example slow-motion video of operating spring-damped underactuated swashplateless rotor, modulated at 270-degrees phase angle, 15-degrees tilt angle.

**Author Contributions:** Conceptualization, H.G.; Methodology, H.G.; Software, H.G.; Validation, H.G.; Formal analysis, H.G.; Investigation, H.G.; Resources, K.C.W.; Data curation, H.G.; Writing—original draft, H.G.; Writing—review & editing, K.C.W.; Supervision, K.C.W.; Project administration, K.C.W.; Funding acquisition, K.C.W. All authors have read and agreed to the published version of the manuscript.

**Funding:** This research received no external funding.

**Data Availability Statement:** Data are contained within the article and supplementary materials.

**Acknowledgments:** We appreciate the help from Pengfei Yu at the USYD UAVLab for his support in MCU electronics hardware troubleshooting during the early stage prototyping of the platform. Also, we would like to thank Duncan Stenger from the School of AMME of the University of Sydney who helped with the customized computer numeric controlled (CNC) manufacturing of the rotor hub mechanism components.

**Conflicts of Interest:** The authors declare no conflicts of interest.

## Abbreviations

The following abbreviations are used in this manuscript:

| | |
|---|---|
| UAV(s) | Unmanned aerial vehicle(s) |
| MAV(s) | Micro aerial vehicle(s) |
| eVToL | Electric vertical take-off and landing |
| GCS | Ground control station |
| MCU | Microcontroller unit |
| ESC | Electronic speed control |
| MAVLink | Micro Air Vehicle Link |
| PWM | Pulse-width modulation |
| SPI | Serial peripheral interface |
| BEM | Blade element method |
| BEMT | Blade element momentum theory |
| CW/CCW | Clock-wise / counter-clock-wise |
| FFT | Fast Fourier transform |

## Appendix A

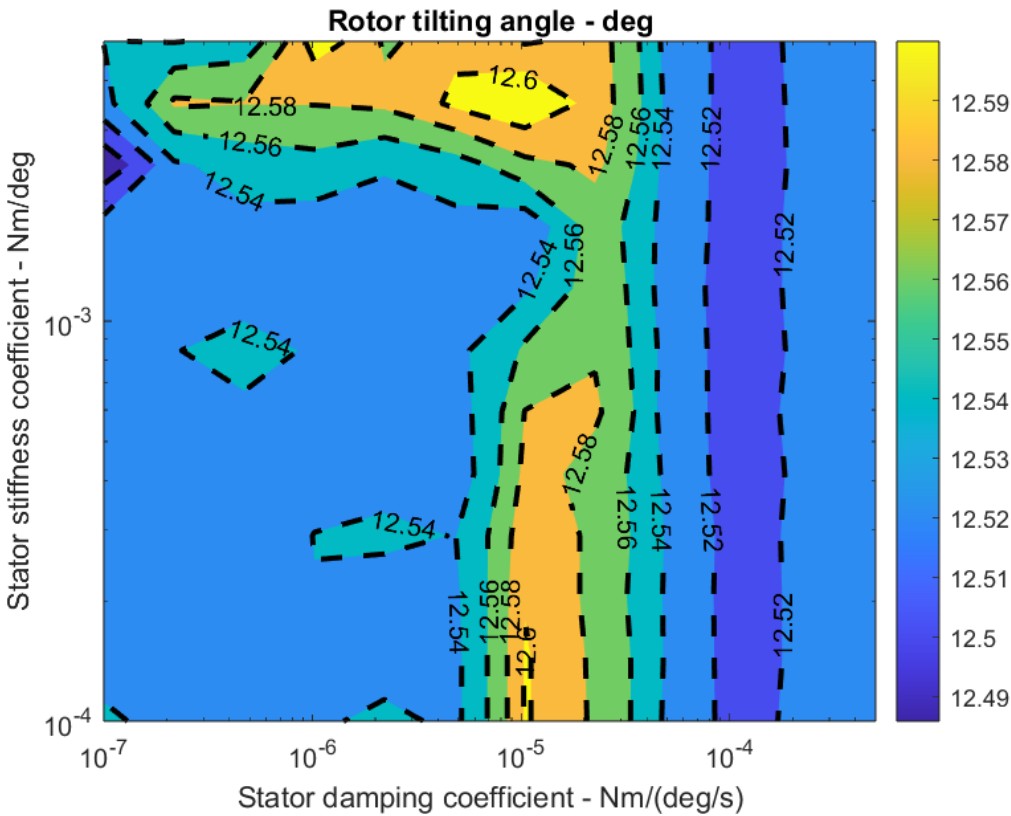

**Figure A1.** Rotor bank angle under current profile.

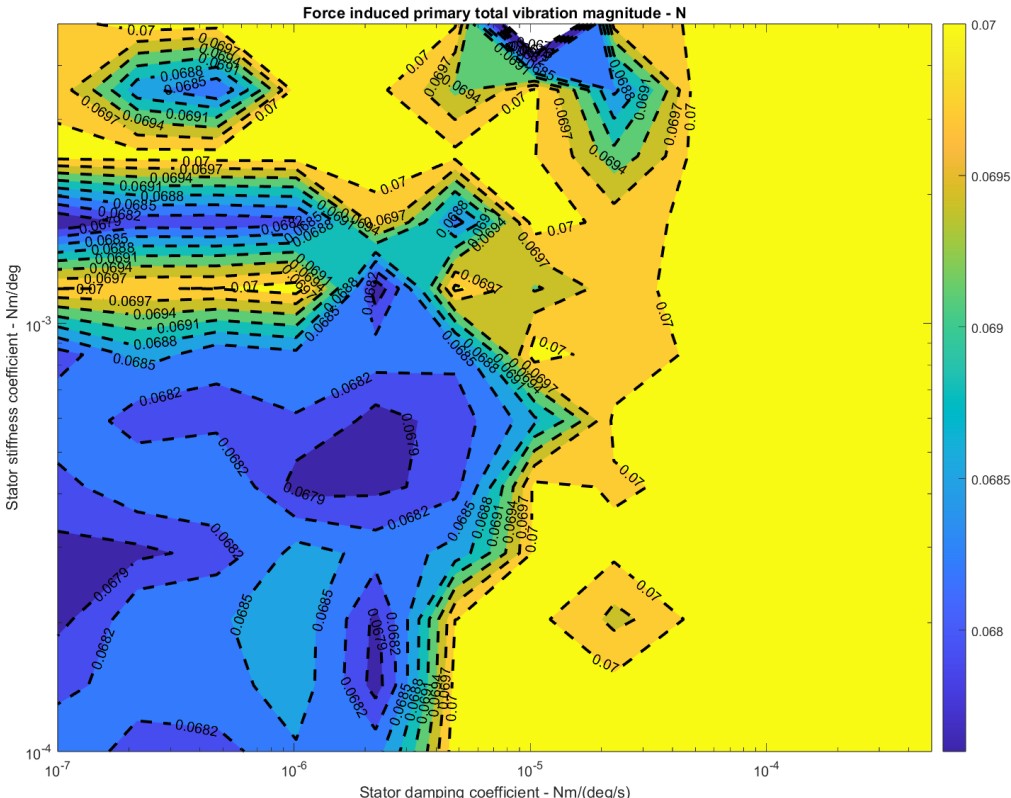

**Figure A2.** Force vector induced peak vibration spectrum amplitude.

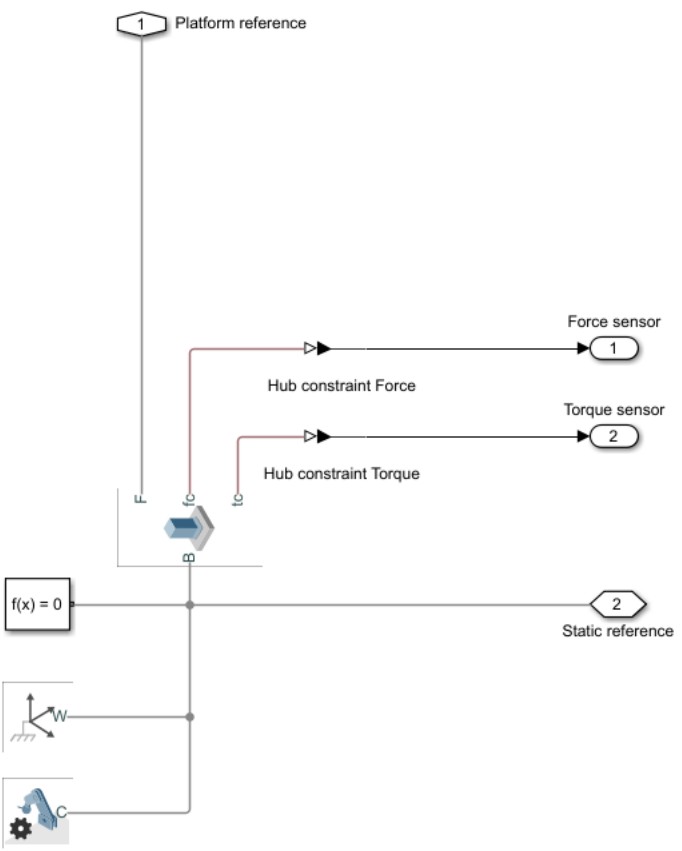

**Figure A3.** Earth reference block of the system.

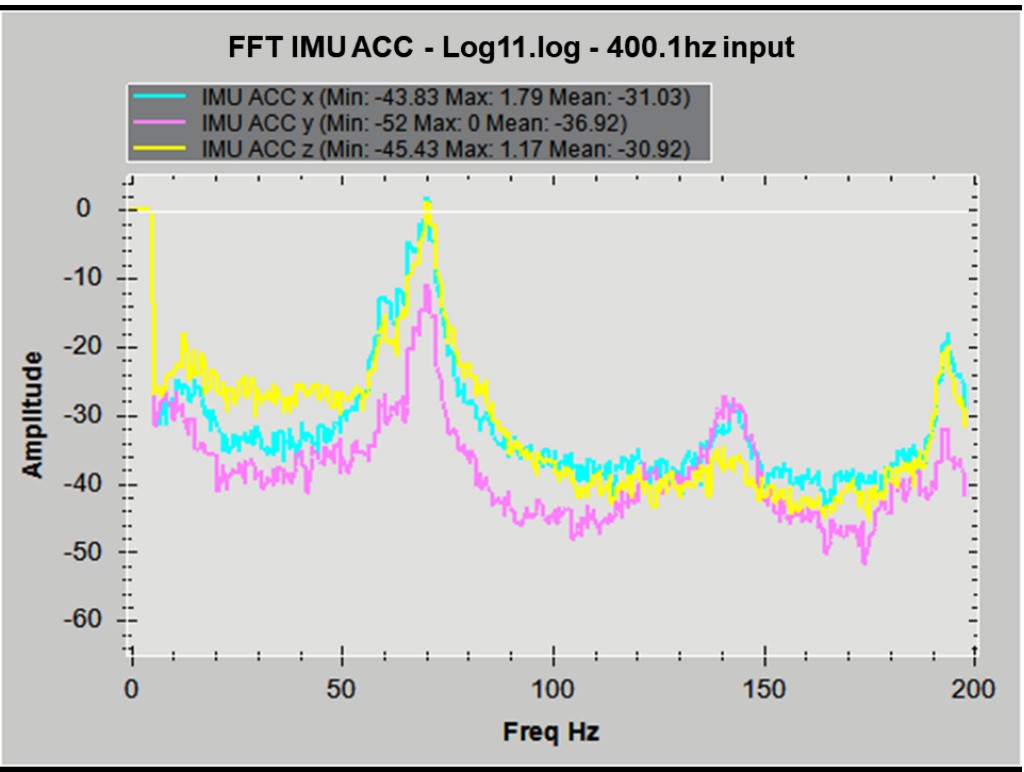

**Figure A4.** One meter high hover flight IMU acceleration signal fast fourier transform with tightened structures on the platform.

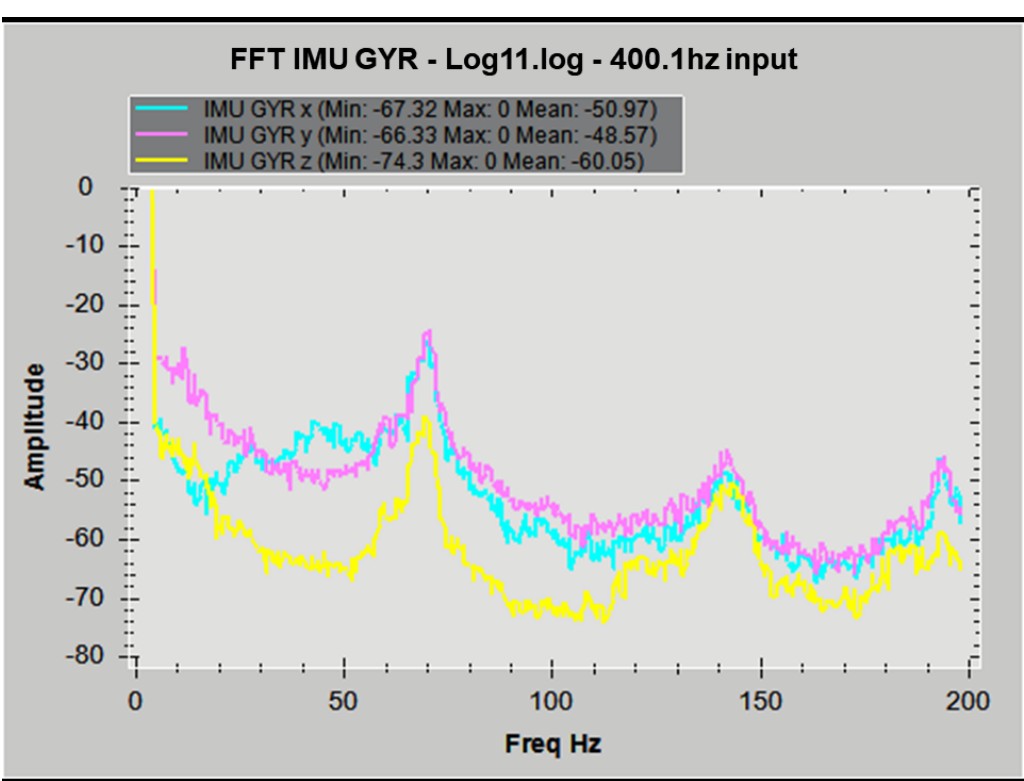

**Figure A5.** One meter high hover flight IMU gyroscope signal fast Fourier transform with tightened structures on the platform.

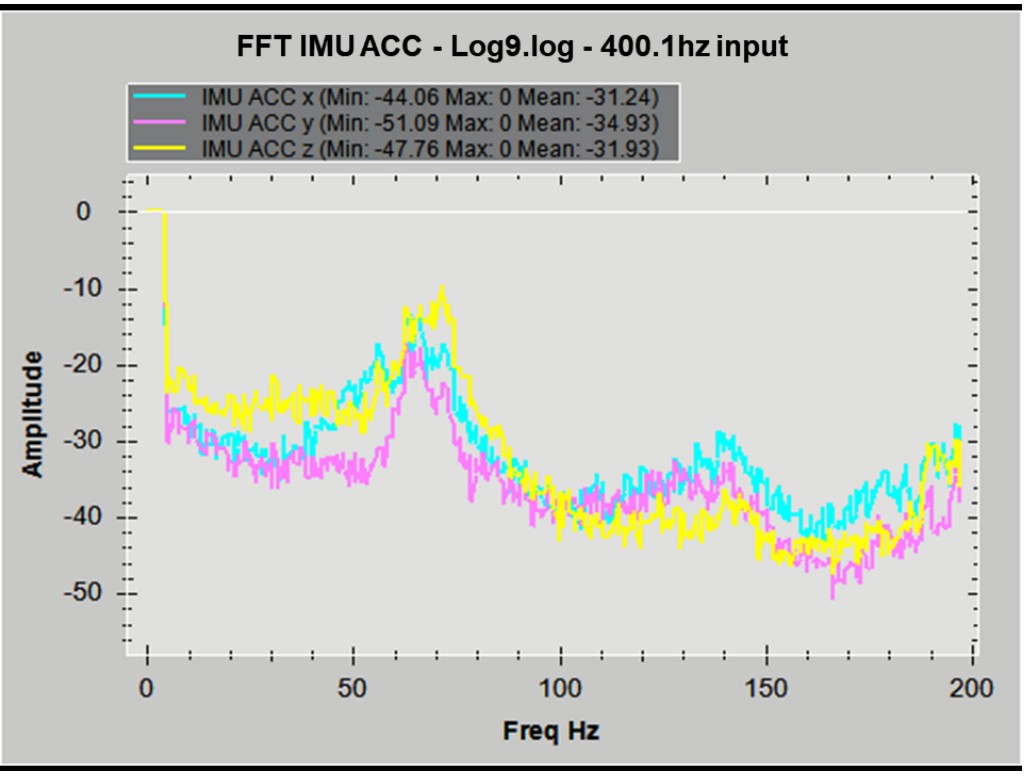

**Figure A6.** One meter high hover flight IMU acceleration signal fast Fourier transform with $7.9 \times 10^{-7}$ Nm/deg spring coefficient in spring-damping structures on the platform.

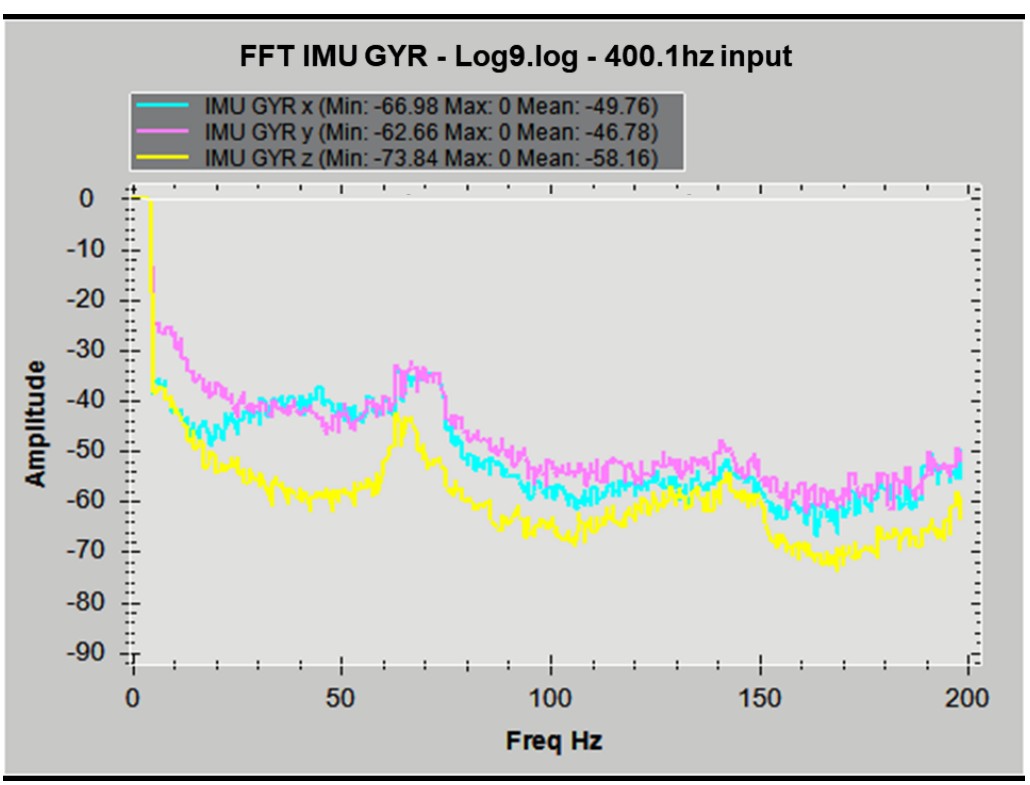

**Figure A7.** One meter high hover flight IMU gyroscope signal fast Fourier transform with $7.9 \times 10^{-7}$ Nm/deg spring coefficient in spring-damping structures on the platform.

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
