# Peer review of "Spring-Damped Underactuated Swashplateless Rotor on a Bicopter Unmanned Aerial Vehicle"

_machines, doi:10.3390/machines12050296_

Round 1

Reviewer 1 Report

Comments and Suggestions for Authors

It is reasoned that a highly efficient spring-damping structure is physically difficult to construct. So, It is desirable that, based on experiments, the authors indicate the some variants the optimal values of spring-damping design, which allows to reduce the level of vibration and improve the quality of controllability of drones.

It is necessary to cite a couple of works of the authors confirming the given researches and experiments.

It is expedient to introduce the proposed spring-damping design into the UAV production as soon as possible.

Reviewer 2 Report

Comments and Suggestions for Authors

The author compares the experimental results of motor angular velocity and vibration magnitude with various spring-damping coefffcients in the shaft axis, the results provided fundamental theories for the system design of vibration isolated the underactuated thrust vectoring rotor to minimize the vibration received by onboard avionics. 

But there are still some issues needed to be deal with.

1. The introduction is too shorter and can not sufficient describles the researched problem and its significance.

2.The refefences needed to be updated so that there are more highly related

documents to support your work.

3.The test example for illustrating ability of dealing with paprameter uncertainty of this UAV should be added into your paper and clarify it.

4.In concclusion,you should added numeric index to prove your results and conclusions.In additionlly, the significance of your work towards future research and application should be explained into article.

5.Some English errors should be cheched and corrected.

Comments on the Quality of English Language

Some English errors should be cheched and corrected.

Reviewer 3 Report

Comments and Suggestions for Authors

Reviewer 4 Report

Comments and Suggestions for Authors

This paper presents a spring-damped swashplateless rotor design on bicopter UAVs. While the design is interesting, it needs further clarification regarding its contributions and effectiveness.

  1. It would be beneficial to include a thorough literature review of the state-of-the-art design and its challenges in the introduction.
  2. What are the major contributions of this work? Please state them clearly.
  3. In Line 49, the authors stated that the system is based on the Kakute F4 mini Ardupilot flight controller. Since Ardupilot firmware is open-source, what are this work's real contributions?
  4. While the authors provided several equations regarding the dynamics, please include the necessary stability analysis (or proof) of the system.
  5. The title of Section 3.4 is "Experiment setup." However, a lot of the content in this manuscript is about simulation in MATLAB. So, is it supposed to be simulation or experiment? Please clearly include the simulation part and experiment part.
  6. While hardware design is presented, why still include a lot of the content in the simulation environment? Does it provide more information that experiments cannot provide?
  7. Please provide a video of the experiment with the hardware designed and developed. It can better illustrate how the design works.

Reviewer 5 Report

Comments and Suggestions for Authors

This is an interesting paper and investigates the implementation of passive damping materials for vibration reduction in underactuated, swashplateless UAVs. Through a combination of simulations and ground tests, the study demonstrates the effectiveness of passive damping in improving UAV performance and reliability by mitigating vibration-induced issues. The findings suggest that integrating passive damping materials into underactuated UAVs can lead to smoother hovering operations and enhanced overall stability, thereby unlocking their full potential for various applications. Overall, it looks good. However, the paper needs some additional inputs to be ready for publication.

Point 1.

Expand the introduction and provide more context on the existing challenges and limitations in this area. For example, mention the lack of comprehensive studies focusing on passive damping materials in underactuated UAVs. It would also be interesting to include a paragraph and discuss the importance of tilt-rotor concepts for small eVTOL aircraft design particularly for the purpose of advanced air mobility (AAM) and urban air mobility (UAM). You may refer to the following publications on the development of new eVTOL aircraft concepts.

1.       Hartmann P, Schütt M, and Moormann D. (2017). Control of departure and approach maneuvers of tiltwing VTOL aircraft. In AIAA Guidance, Navigation, and Control Conference (p. 1914). https://doi.org/10.2514/6.2017-1914

2.       Rostami M, Bardin J, Neufeld D, Chung J. EVTOL Tilt-Wing Aircraft Design under Uncertainty Using a Multidisciplinary Possibilistic Approach. Aerospace. 2023; 10(8):718. https://doi.org/10.3390/aerospace10080718

3.       Misra A, Jayachandran S, Kenche S, Katoch A, Suresh A, Gundabattini E, Selvaraj S K, and Legesse A A. 2022. A review on vertical take-off and landing (VTOL) tilt-rotor and tilt wing unmanned aerial vehicles (UAVs). Journal of Engineering, 2022. https://doi.org/10.1155/2022/1803638

Point 2.

For figures 15, 16, 17, 19 and 20, increase the font size of the axis.

Point 3.

In the results section, discuss the significance of the rotor mechanism damping coefficient estimation and the implications for vibration reduction in rotor systems.

Point 4.

Expand the discussion on the practical implications of the findings. Address how the proposed spring-damping structures could be applied in real-world scenarios to mitigate vibration and improve the performance of thrust vectoring rotor platforms. You might refer to the potential benefits of the eVTOL aircraft concepts here as well.

Comments on the Quality of English Language

Minor editing of English language required

Round 2

Reviewer 3 Report

Comments and Suggestions for Authors

I have no more comments. 

Reviewer 4 Report

Comments and Suggestions for Authors

The authors have addressed all my previous comments properly. I have no further questions.

Reviewer 5 Report

Comments and Suggestions for Authors

The authors stressed my concerns.

Comments on the Quality of English Language

Minor editing of English language is required